# Resistance profile of the HIV-1 maturation inhibitor GSK3532795 in vitro and in a clinical study

Ira Dicker[1], Sharon Zhang[1], Neelanjana Ray[2], Brett R. Beno[3], Alicia Regueiro-Ren [4], Samit Joshi[5], Mark Cockett[1], Mark Krystal[1]*, Max Lataillade[5]

**1** Department of HIV Discovery, ViiV Healthcare, Branford, Connecticut, United States of America, **2** Department of Early Development, Bristol-Myers Squibb Research and Development, Princeton, New Jersey, United States of America, **3** Department of Molecular Discovery Technologies, Bristol-Myers Squibb Research and Development, Wallingford, Connecticut, United States of America, **4** Department of Chemistry Bristol-Myers Squibb Research and Development, Wallingford Connecticut, United States of America, **5** Department of Early Development, ViiV Healthcare, Branford, Connecticut, United States of America

* Mark.R.Krystal@viivhealthcare.com

## Abstract

GSK3532795 (formerly BMS955176) is a second-generation maturation inhibitor (MI) that progressed through a Phase 2b study for treatment of HIV-1 infection. Resistance development to GSK3532795 was evaluated through *in vitro* methods and was correlated with information obtained in a Phase 2a proof-of-concept study in HIV-1 infected participants. Both low and high concentrations of GSK3532795 were used for selections *in vitro*, and reduced susceptibility to GSK3532795 mapped specifically to amino acids near the capsid/ spacer peptide 1 (SP1) junction, the cleavage of which is blocked by MIs. Two key substitutions, A364V or V362I, were selected, the latter requiring secondary substitutions to reduce susceptibility to GSK3532795. Three main types of secondary substitutions were observed, none of which reduced GSK3532795 susceptibility in isolation. The first type was in the capsid C-terminal domain and downstream SP1 region (including (Gag numbering) R286K, A326T, T332S/N, I333V and V370A/M). The second, was an R41G substitution in viral protease that occurred with V362I. The third was seen in the capsid N-terminal domain, within the cyclophilin A binding domain (V218A/M, H219Q and G221E). H219Q increased viral replication capacity and reduced susceptibility of poorly growing viruses. In the Phase 2a study, a subset of these substitutions was also observed at baseline and some were selected following GSK35323795 treatment in HIV-1-infected participants.

## Introduction

HIV-1 assembly initiates when the viral Gag polyprotein begins to multimerize at the plasma membrane of an infected host cell [1]. As the virus buds, the viral protease (Pr) cleaves between each of the major domains of Gag (matrix [MA], capsid [CA] and nucleocapsid [NC]), resulting in major structural changes that evolve an immature noninfectious virus particle into a mature, infectious virion. This process is collectively termed "maturation" and is crucial for

**Funding:** Funding for all studies described came directly from Bristol-Myers Squibb. The funder provided support in the form of salaries for all authors but did not have any additional role in the study design, data collection and analysis, decision to publish, or preparation of the manuscript. The specific roles of these authors are articulated in the 'author contributions' section.

**Competing interests:** Although the authors were all employees of BMS at the time of the study, this affiliation does not alter the authors' adherence to PLOS ONE policies on sharing data and materials. Rights to the compound were subsequently transferred to ViiV Healthcare.

viral infectivity. Spacer peptide 1 (SP1), a sub-domain within the Gag polyprotein, plays an essential role in stabilizing the immature HIV-1 Gag lattice [2]. Cleavage between CA and SP1 is the final step in the maturation process that triggers structural rearrangements leading to the condensed conical core characteristic of a fully infectious viral particle. Mutating HIV-1 Gag adjacent to the CA/SP1 boundary results in the production of immature, noninfectious particles [3, 4]. Thus, pharmacological intervention at this site provides a basis for new anti-HIV agents. A prior maturation inhibitor (MI), bevirimat, demonstrated dose-dependent antiviral activity in clinical studies [5]. However, further development was suspended after a better understanding of resistance to the molecule was obtained [6–8] and significantly reduced bevirimat susceptibility was associated with naturally occurring polymorphisms at Gag amino acids 369, 370 and 371, found in ~50% of patient isolates [9, 10] (**Fig 1**). Subsequent reports identified Gag V362I (15.4% of the subtype B Los Alamos National Labs [LANL] database)[11] as another major polymorphic variant conferring reduced bevirimat susceptibility [12]. Substitutions in the CA C-terminal domain (CA-CTD) were also observed during *in vitro* selections using another MI of unrelated structure [13, 14].

GSK3532795 (formerly BMS-955176) is a second-generation MI with broad *in vitro* activity against HIV-1 and enhanced activity against certain bevirimat resistant variants [15–18]. In a Phase 2b study (GSK 205891, previously AI468038), GSK3732795 exhibited efficacy comparable with that of efavirenz (EFV). However, higher rates of gastrointestinal intolerability and treatment-emergent resistance to the nucleoside reverse transcriptase inhibitor (NRTI) backbone relative to EFV prevented further development and discontinuation of GSK3532795 [19]. Here we report on the *in vitro* and clinical genotypic resistance profile of GSK3532795; clinical data were obtained from the Phase 2a clinical study (AI468002) [20].

## Materials and methods

### Compounds

GSK3532795 (BMS-955176) was synthesized at Bristol-Myers Squibb (BMS).

### Virus and cells

MT-2 cells and the proviral DNA clone NL4-3 were obtained from the NIH AIDS Research and Reference Reagent Program. MT-2 cells were propagated at 37˚C/5% $CO_2$ in RPMI 1640 media (Gibco) supplemented with 10% heat inactivated fetal bovine serum (FBS, Gibco), 100 U/mL of penicillin and 100 μg/mL of streptomycin (Gibco), and sub-cultured twice a week. Virus stocks used to initiate selection were generated by transfecting 293T cells (Lipofectamine PLUS kit, Invitrogen) with proviral DNA clones of NL4-3 Gag P373S (hereafter referred to as wild-type) and NL4-3 Gag P373S with additional defined Gag amino acid substitutions introduced by site-directed mutagenesis. The Gag P373S substitution was included to better represent the subtype B clinical population: S373 is near the SP1 cleavage site and is present in 60% of subtype B isolates [11]. Luciferase reporter variants of NL4-3 (RepRluc Gag P373S) contained the *Renilla* luciferase (Rluc) gene in the *nef* locus as previously described [15].

### Drug susceptibility assay

Multiple-cycle drug susceptibility assays were carried out as previously described [15]. Briefly, MT-2 cells were infected with virus at a multiplicity of infection (MOI) of 0.005. Cell-virus mixtures were seeded onto 96-well plates containing serially diluted compound at a final density of 10,000 cells per well. After 4–5 days of incubation, virus yield was quantified by either cell-free reverse transcriptase (RT) activity (scintillation proximity assay [SPA]), or cell-

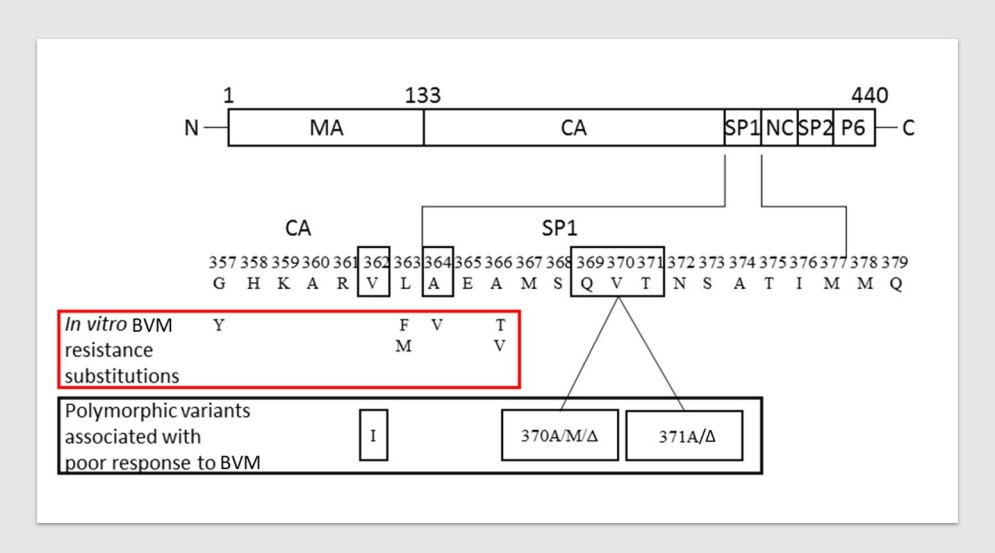

**Fig 1. Protease processed HIV-1 Gag polyprotein and amino acid differences that contribute to reduced bevirimat (BVM) susceptibility.** (Top): Gag region from HIV-1 showing segments of the structural proteins cleaved by HIV Pr. Bottom: Blowup of region surrounding SP1; amino acids observed as substitutions resulting in reduced susceptibility to BVM *in vitro* are shown in the red box. Polymorphic amino acid variations within and near SP1 that reduce BVM susceptibility are shown in the grey box. Figure adapted from [12].

associated *Renilla* luciferase activity (Dual-Luciferase® Reporter Assay System, Promega). Site directed mutant viruses were assayed using the RT endpoint. The 50% effective concentration (EC50) was derived from plots of percent inhibition of luciferase or RT activity versus $\log_{10}$ drug concentration. Each experiment contained triplicate wells for each virus tested and the $EC_{50}$ was calculated as an average of the set. Standard deviations were determined using data from separate experiments.

### *In vitro* selections starting at high compound concentration

Virus breakthrough experiments in the presence of GSK3532795 were performed at a fixed concentration of 30 x EC50 of compound for each virus, with initial infection at low MOI (0.005) or high MOI (0.05). Every 3–4 days, cultures were split 1:3 into fresh media with the same concentration of compound. Separate cultures with no compound added were used as a control. Virus breakthrough was considered to have occurred when 100% cytopathic effect (CPE) was observed. At this point cultures were terminated, and cell supernatant were harvested for population sequencing of Gag genes.

### *In vitro* selections starting at low compound concentrations

Viruses with decreased susceptibility to GSK3532795 were selected by serial passage of viral supernatants onto fresh cells in the presence of increasing concentrations of compound. Selections were started with 2 x 106 MT-2 cells infected with virus at a MOI of 0.005 and cultured at 2 x 105 cells/ml in the presence of GSK3532795 at 1x or 2x the $EC_{50}$ for the starting virus variant. Infected MT-2 cells without compound were passaged in parallel (no drug controls). At the end of each passage (100% CPE) 25 µL of culture supernatant was transferred into a fresh cell culture (10 mL, 2 x 105 cells per mL) with a 2-fold increased drug concentration. Selections were terminated after passage 8 when compound cytotoxicity was observed.

## Replication capacity assays

MT-2 cells (0.1 x 106 cells/mL) were infected with RepRluc Gag P373S or reporter-free Gag variants at a MOI of 0.01. Infected cells were seeded into triplicate wells of seven 96-well plates and incubated for up to 7 days. Starting from day 0, cells and supernatants from one plate were harvested each day and stored at -80°C for subsequent analysis. Virus yields were assessed by RT SPA, p24 ELISA (PerkinElmer, NEK050B001KT Alliance HIV-1 P24 ANTIGEN ELISA Kit, 480 Test) or by luciferase activity (for RepRluc cultures). Viral growth rates were calculated from slope of plots of log RT versus time. Replication capacity (RC) was defined as the growth rate for the Gag variant relative to that of wild-type virus (with or without luciferase gene), expressed as a percentage.

## Population sequencing

Viral DNA was isolated from infected cells (Blood & Cell Culture DNA Midi kit, Qiagen), and amplified by PCR with primers specific for the desired Gag/Pr region of the genome (5′-GA CTCGGCTTGCTGAAGCGCGCACGGCAAGAGGCGAGGGGCGGCG-3′ and 5′-CAGGCCCA ATTTTTGAAATTTTTCC-3′). Amplification was performed for 35 cycles using Platinum Taq (Invitrogen). PCR products were purified using the QIAquick PCR purification kit (Qiagen). Sequencing of purified Gag/Pr PCR products was performed at the BMS Core Sequencing Facility and data was analyzed using Sequencher software, version 4.6.

## Structural modeling of capsid and compound

A model of GSK3532795 bound to an immature CA/SP1 hexamer with a partial SP1 6-helix bundle was created based on the cryo-electron microscopy structure of immature CA/SP1 (Gag residues 148–371; Protein Data Bank ID: 5L93) which contains the complete CA-NTD and CA-CTD along with 8 residues of the SP1 region [21]. An initial CA/SP1 hexamer model was generated from the published coordinates for the cryo-electron microscopy model using the Maestro molecular modeling software suite [22]. The Schrödinger Protein Preparation Wizard was then used to add hydrogen atoms to the model and optimize hydrogen-bonding interactions [23, 24]. Following this, three stages of energy minimization were performed on the model with MacroModel, the OPLS3e molecular mechanics force field, an implicit water solvation model, and a 0.05 kJ/Å·Mol gradient convergence threshold [25, 26]. In the first stage, the positions of all non-hydrogen atoms were fixed and 500 steps of PRCG minimization were performed. In the second stage, only the positions of protein backbone atoms were fixed, and 100 steps of steepest descent minimization were performed. Finally, all constraints were removed, and the model was subjected to 500 steps of PRCG minimization. A model of HIV maturation inhibitor GSK3532795 was also created in Maestro and missing/low-quality torsion parameters were generated in the context of the OPLS3e force field using the Maestro Force Field Builder. Following energy minimization of the GSK3532795 model, attempts to dock the ligand into the CA/SP1 hexamer model did not yield any predicted binding poses within the SP1 six-helix bundle, which is where the HIV maturation inhibitor bevirimat is predicted to bind based on cryo-EM data [21]. Thus, GSK3532795 was manually modeled into the center of the SP1 six-helix bundle with the carboxylate moiety oriented toward the ring of six Lys359 residues in the CA/SP1 hexamer model and the isoprenyl moiety proximal to Met367 in analogy to the binding mode proposed for bevirimat [27]. The model was subjected to 500 steps of energy minimization in MacroModel with the PRCG algorithm, OPLS3e force field, and implicit water solvation with the positions of all protein backbone atoms fixed. The resulting model was used as input for a fully-solvated (TIP3P water model) 25 ns molecular dynamics simulation (NPT ensemble) using the OPLS3e force field with Desmond and default

model relaxation and equilibration protocols [28, 29]. In the molecular dynamics study, harmonic restraints of 1.0 kcal/mol·Å$^2$ were applied to all protein backbone atoms. A software tool developed in-house was used to identify the molecular dynamics snapshot closest to the average structure for the final 1 ns of the simulation and this was used for figure generation.

## GSK3532795 Phase 2a clinical study

AI468002 (NCT01803074) was a Phase 2a, randomized, dose-ranging multipart study that investigated GSK3532795 in HIV-1 subtype B- and C-infected individuals [20]. In part A of the study, HIV-1 subtype B-infected individuals received 5–120 mg GSK3532795 (or placebo) once daily (QD, quaque die) for 10 days. Participants with a history of genotypic/phenotypic drug resistance to protease inhibitors (PIs) or with HIV-1 genotypic drug resistance to PIs at baseline (including D30N, M46I/L, I47V/A, G48V, I50L, I54M/L, Q58E, T74P, L76V, V82A/F/L/T/S, N83D, I84V, N88S, or L90M) were excluded, despite any reported effects of PI resistance or resistance markers on GSK3532795 susceptibility (15). The impact of baseline Gag polymorphisms on response to GSK3532795 monotherapy was assessed for participants in part A of the study (N = 60).

HIV genotypic and phenotypic resistance analysis was carried out for all participants at baseline and post-treatment (day 10, 10–120 mg arms) by Monogram Biosciences, as previously described [30]. Gag/Pr regions were amplified at Monogram Biosciences from plasma by RT PCR. Amplicons were sequenced using population sequencing methods and were used for the assessment of GSK3532795 susceptibility in the PhenoSense HIV-1 Gag/Pr assay [31, 32].

## Results

### Selection of viruses with reduced susceptibility to GSK3532795 *in vitro*

The selection of viruses with reduced sensitivity to GSK3532795 *in vitro* was examined using two different approaches. In the first approach, NL$_{4-3}$ Gag P373S virus was sequentially passaged beginning at a low concentration of GSK3532795 (either 1x or 2xEC$_{50}$). Infected cells were carried until significant CPE was observed and then the supernatant was used to infect new cells and the concentration of GSK3532795 was doubled. Two types of wild type NL$_{4-3}$ Gag P373S virus were used; one was taken straight from the freezer and the other was first passaged 12 times in the absence of drug in order to increase the potential heterogeneity within the genome. Multiple independent selections were performed for each virus, and variants with reduced susceptibility to GSK3532795 were selected in wild-type virus by passage 8 in two direct virus and two culture-adapted virus passages in the presence of GSK3532795. Selected Gag and Pr substitutions are shown in **Table 1**. In three cultures, A364V became the dominant population and was selected with other Gag mutations (G221E, V159I, V218I, R286K and V362I). In the other culture, V362I was selected with other substitutions in Gag (A118T, V218M, T332S) and Pr (R41G). These viruses were all resistant to GSK3532795 with EC$_{50}$ values > 0.25μM. All viruses remained sensitive to the PI, nelfinavir (NFV). In both the drug-free control passages, V218M emerged as the dominant population within 8 passages (not shown), suggesting this mutation is selected based upon improved fitness in cell culture, irrespective of MI treatment.

A similar selection experiment was performed with NL$_{4-3}$ Gag P373S virus (and a 12 passage population) that also contained V370A, a substitution shown to affect bevirimat susceptibility [9, 10]. For the V370A-containing virus, sequences were obtained at each passage and treatment with increasing concentrations of GSK3532795 resulted in sequential accumulation of Gag substitutions (**Table 2**). All viruses maintained the V370A site-directed mutation throughout the selection. At passage 1 (after 13 days) V362I comprised 30% of the population.

By passage 2 (35 days), this substitution was lost and stably replaced by A364V (100% of the population). A364V was carried through the remaining passages, conferring >200-fold decreased susceptibility to GSK3532795. The EC50 values for passages 2 to 10 ranged from 1–2 μM (Table 2). Additional emergent substitutions included two changes in the SP2/p6 cleavage site: P453S, co-selected at 100% with A364V after passage 2, and S495N that emerged by day 78 but did not increase beyond 20–30% over the course of 5 passages (45 days). These substitutions have been cited as accessory mutations linked to PI susceptibility and fitness [33]. V218M/A, H219Q, T239I in the CA N-terminal domain (CA-NTD) were also observed; T239I was present only transiently but the other two changes were fixed by the last passage. Finally, two additional substitutions emerged in the MA region: I34V appeared at passage 3 at 30% and gradually increased to 70%, until the selection was terminated on day 113, while V35I appeared transiently (~20%) between passages 7 and 9 and was then lost. I34V has been described in a patient during PI therapy as a compensatory mutation [33], and may be acting in a similar capacity for GSK3532795. As with the wild-type virus cultures, V218M emerged after 12 passages of V370A-containing virus in the absence of drug. A second drug-free control culture acquired the adjacent change of H219Q. In the passages initiated with culture-adapted (p12) V370A-containing virus, a different pattern of substitutions was seen, with V362I dominating at passage 8 in both cultures, in one case accompanied by K114V in the MA and K14E in Pr, and in the other case by N235T (present at 50%) (Table 2). In addition to these changes, the culture-adapted V370A-containing virus already had V218M as the majority population and this was maintained through all subsequent passages with or without GSK3532795.

Although GSK3532795 inhibits most of the Gag polymorphic viruses that are less susceptible to the first-generation MI bevirimat,[15] the prevalence of these polymorphisms in the clinic suggested a need to further understand the *in vitro* potential for the selection of GSK3532795 resistance starting with these polymorphic variants. Thus, additional resistance selection experiments were performed with an expanded panel of Gag polymorphic variant viruses, including the single variants V362I, R286K, T332S, ΔV370 or ΔT371 and double variants V362I/V370A, R286K/V370A or T332S/V362I. Although resistance was selected with all variants, no new substitutions emerged during selection beyond those previously identified (S1 Table).

**Table 1. Genotypic and phenotypic changes selected by serial passage of wild-type virus with GSK3532795.**

| Selection | | | | Substitutions, % | | | | | | | | | | EC$_{50, \mu}$M (FC) | |
|---|---|---|---|---|---|---|---|---|---|---|---|---|---|---|---|---|
| Virus* | P # | Days | Final Conc. (nM) | MA | | | CA | | | | CA/SP1 | | Pr | GSK 3532795 | NFV |
| | | | | A | A | V | V | G | R | T | V | A | R | | |
| | | | | 118 | 119 | 159 | 218 | 221 | 286 | 332 | 362 | 364 | 41 | | |
| | | | | T | V | I | M | E | K | S | I | V | G | | |
| WT | 0 | - | 0 | - | - | - | - | - | - | - | - | - | - | 0.002 | 0.004 |
| | 8 | 63 | 0 | - | - | - | 100 | - | - | - | - | - | - | 0.001 (<1) | 0.002 (<1) |
| | 8 | 97 | 256 | 60 | - | - | 100 | - | - | 100 | 100 | - | 100 | 0.257 (128) | 0.001 (<1) |
| | 8 | 83 | 256 | - | - | - | - | 100 | - | - | - | 100 | - | 2.3 (>1000) | 0.002 (<1) |
| WT p12 | 0 | - | 0 | - | - | - | - | - | - | - | - | - | - | nd | nd |
| | 8 | 63 | 0 | - | 20 | - | 100 | - | - | - | - | - | - | 0.001 (<1) | 0.002 (<1) |
| | 8 | 70 | 256 | - | - | - | 100 | - | - | - | 10 | 100 | - | 0.43 (21) | 0.001 (<1) |
| | 8 | 70 | 256 | - | - | 20 | 100 | - | 40 | - | 20 | 90 | - | 0.88 (44) | 0.002 (<1) |

CA, capsid; EC50, 50% effective concentration; FC, fold change relative to starting virus; MA, matrix; nd, not done; NFV, nelfinavir; P#, passage number; Pr, protease; SP1, spacer peptide 1; WT, wild-type

*NL$_{4-3}$ Gag P373S reporter-free virus, without (WT) or with (WTp12) prior culture adaptation

**Table 2. Genotypic and phenotypic changes selected by serial passage of V370A virus with GSK3532795.**

| | | | | MA | | | CA | | | | CA/SP1 | | SP2/p6 | | PR | EC50, µM (FC) | |
|---|---|---|---|---|---|---|---|---|---|---|---|---|---|---|---|---|---|
| Virus* | P# | Days | Conc (nM) | I 34 V | V 35 I | K 114 V | V 218 M/A | H 219 Q | T 239 I | N 235 T | V 362 I | A 364 V | P 453 S | S 495 N | K 14 E | GSK 3532795 | NFV |
| **V370A** | 0 | 0 | 0 | - | - | - | - | - | - | - | - | - | - | - | - | 0.004 | 0.005 |
| | 12[†] | 97 | 0 | - | - | - | - | 100 | - | - | - | - | - | - | - | 0.003 (<1) | 0.002 (<1) |
| | 12[‡] | 97 | 0 | - | - | - | 100 | - | - | - | - | - | - | - | - | 0.001 (<1) | 0.001 (<1) |
| | 1 | 13 | 8 | - | - | - | - | - | - | - | 30 | - | - | - | - | nd | nd |
| | 2 | 35 | 16 | - | - | - | - | - | - | - | 100 | 100 | - | - | - | 1.5 (>250) | 0.001 (<1) |
| | 3 | 47 | 32 | 30 | - | - | 70 | - | - | - | 100 | 100 | - | - | - | 1.1 (>250) | 0.002 (<1) |
| | 4 | 54 | 64 | 45 | - | - | 80 | - | - | - | 100 | 100 | - | - | - | 1.8 (>250) | 0.001 (<1) |
| | 5 | 68 | 128 | 50 | - | - | 95 | - | - | - | 100 | 100 | - | - | - | 1.0 (250) | 0.0002 (<1) |
| | 6 | 78 | 256 | 70 | - | - | 95 | - | - | - | 100 | 100 | 30 | - | - | 1.1 (>250) | 0.001 (<1) |
| | 7 | 84 | 512 | 70 | 20 | - | 90 | - | 20 | - | 100 | 100 | 30 | - | - | 1.2 (>250) | 0.002 (<1) |
| | 8 | 89 | 1024 | 70 | 20 | - | 95 | - | 20 | - | 100 | 100 | 20 | - | - | 1.8 (>250) | 0.003 (<1) |
| | 9 | 95 | 2048 | 70 | 10 | - | 95 | - | - | - | 100 | 100 | 20 | - | - | 1.5 (>250) | 0.003 (<1) |
| | 10 | 106 | 4096 | 70 | | - | 95 | - | - | - | 100 | 100 | 20 | - | - | 1.2 (>250) | 0.001 (<1) |
| **V370A p12** | 0 | 0 | 0 | - | - | - | 100 | - | - | - | - | - | - | - | - | nd | nd |
| | 8 | 63 | 0 | - | - | - | 100 | - | - | - | - | - | - | - | - | 0.004 (1) | 0.005 (1) |
| | 8[a] | 97 | 512 | - | - | 100 | 100 | - | - | - | 100 | - | - | - | 100 | 0.25 (6.25) | 0.001 (<1) |
| | 8[a] | 81 | 512 | - | - | - | 100 | - | - | 50 | 100 | - | - | - | - | 0.35 (8.75) | 0.004 (<1) |

CA, capsid; EC50, 50% effective concentration; FC, fold change relative to starting virus; MA, matrix; NFV, nelfinavir; P#, passage number; SP1, spacer peptide 1; SP2, spacer peptide 2

[†] Control culture 1.

[‡] Control culture 2

[a]: independent virus passages

*NL4-3 Gag P373S reporter-free virus containing V370A polymorphism

Selection experiments were also carried out using a high fixed dose concentration of GSK3532795 that corresponded to a 30xEC$_{50}$ concentration against the virus used. Here, either WT NL$_{4-3}$ P373S, or the same virus with site-directed mutants V370A and/or ΔT371 were analyzed. In addition, two different MOIs were used in selections, either a MOI of 0.005 or ten-fold higher. Cultures were carried at this compound concentration as described until virus breakthrough occurred and then harvested and population sequenced were performed. Virus breakthrough was not observed in all selections, and only the Gag/Pr genes from viruses with emergent substitutions are shown in **Table 3**. In cultures of wild-type virus treated with GSK3532795, the emergence of I333V was observed, while in cultures of V370A-containing virus emergence of V362I occurred (in two cultures), with mixtures of H219H/Q and I333I/V in another set of V370A cultures. The double site-directed mutant, V370A/ΔT371 selected for A326A/T and V362V/I at breakthrough. As shown in **Table 3**, some virus cultures were passaged a further nine times (labeled FU for follow-up), with a 2-fold increase in drug concentration every 3 passages, to enrich the variant population. In the V370A virus passage, I333I/V and H219H/Q mixtures were consolidated to I333V, H219Q within the first passage; this variant grew steadily for further passages in the presence of increasing GSK532795 concentrations of up to 120xEC50. Starting with the double mutant V370A/ΔT371, initial mixtures of A326A/T and V362V/I were replaced by two new substitutions, H219Q and A364V within three

passages. Thus, regardless of the genotype and method of selection used, there appears to be a limited set of amino acid substitutions selected by GSK3532795.

## Susceptibility of site-directed mutant viruses to GSK3532795

Results from selection experiments and published bevirimat resistance profiles,[9, 10, 12, 34] were used to generate a large family of site-directed mutants in NL4-3 RepRluc. Drug susceptibility of these variants are shown in Table 4. Replication capacity (RC), when available, are also shown. Among single amino acid substitutions and deletions representing major polymorphic variants and selected changes (Group 1, known MI substitutions) the change that engendered the greatest difference in GSK3532795 susceptibility resulted from A364V (>862-fold), which also conferred high level cross-resistance to bevirimat. The most robust viruses in terms of growth were A364V and V370A, while V362I exhibited a lower RC. The deletion at amino acid 370 conferred a modest reduction in GSK3532795 susceptibility but substantially reduced RC to only 20%. Though present at only 1.1% among subtype B isolates, this single deletion was used as a surrogate for a single deletion in the region of 370–372, where there are substantial differences in the percentage of single deletions among subtypes [B (9.6%), C (95.3%), and all non-B (81.3%)] Viruses [11]. In non-B subtypes such single deletions are usually present in the context of V370A, with that change likely increasing fitness. For example, the replication capacity of NL$_{4-3}$ V370A/ΔT371 (a.k.a. ΔV370/T371A) virus was 74% as compared to 20% for ΔV370 (Table 4).

The secondary substitutions that arose in combinations during *in vitro* selections (group 2) in isolation imparted small reductions (0.7–2.5-fold) in GSK3532795 susceptibility. When polymorphic variants V370A, ΔV370 or V362I were added to a subset of these (group 3), GSK3532795 susceptibility decreased 1.9–208 -fold. Many double variants remained relatively sensitive (<12 -fold change), but three viruses stood out with high fold-changes. These are A326T/ΔV370 (29 -FC), R286K/V362I (61 -FC) and V362I/V370A (208 -FC). Where assessed, these double variants with larger reductions in susceptibility exhibited substantial reductions in RCs (41% RC for R286K/V362I and 60% RC for V362I/V370A) (Table 4).

**Table 3. Amino acid substitutions selected in Gag during passage at a high fixed concentration of GSK3532795 (30xEC$_{50}$).**

| Start Virus | MOI | # cultures that gave breakthrough with genotype changes | Days[b] | Gag region[a] | | | | |
|---|---|---|---|---|---|---|---|---|
| | | | | Capsid | | | Capsid/SP1 | |
| | | | | 219 | 326 | 333 | 362 | 364 |
| WT[c] | 0.05[d] | 1/2 | 19 | - | - | I333V | - | - |
| V370A | 0.005 | 1/3 | 25 | - | - | - | V362I | - |
| | 0.05 | 1/4 | 14 | - | - | - | V362I | - |
| V370A | 0.005 | 2/3 | 31 | H219H/Q | - | I333V/I | - | - |
| | 0.05 | 2/3 | FU | H219Q | - | I333V | - | - |
| ΔT371/ V370A | 0.05[d] | 1/2 | 17 | - | A326A/T | - | V362V/I | - |
| | | 1/2 | FU | H219Q | - | - | - | A364V |

[a]There were no changes detected in the matrix or SP2/p6 regions of Gag.

[b]Number of days in culture.

[c]Wild type NL$_{4-3}$. virus

[d] MOI of 0.005 did not produce virus

EC$_{50}$, 50% effective concentration; FU, follow up selection using nine sequential passages at two-fold increases in GSK3532795; MOI, multiplicity of infection; SP1, spacer peptide 1, WT, wild-type

**Table 4. Antiviral sensitivities of site-directed mutants.**

| Group | Genotype | FC EC$_{50}$ | | % in LANL[a] | % in LANL[a] | RC (%) |
|---|---|---|---|---|---|---|
| | | GSK 3532795 | bevirimat | LANL Subtype B | LANL all subtypes | |
| 1 | WT[b] | 1.0 ± 0.5 (1.2 nM) | 1.0 ± 0.2 (12.7 nM) | 50.8 | 39.3 | 100 |
| | V370A | 1.8 ± 1.3 | 110 ± 24 | 16.2 | 39.9 | 100 |
| | V362I | 2.2 ± 1.2 | 0.6 | 15.4 | 7.8 | 68 |
| | ΔT371 | 2.0 ± 0.5 | 28 | 0.06 | 0.02 | 90 |
| | V370A/ΔT371 | 5.5 ± 0.6 | >862 | 0 | 0.06 | 74 |
| | ΔV370 | 5.8 ± 1.3 | >862 | 1.1 | 0.08 | 20 |
| | A364V | 835 | >862 | 0.18 | 0.13 | 82 |
| 2 | T332S | 1.9 ± 0.5 | 21 | 0.37 | 22.9 | 61 |
| | A326T | 2.2 ± 0.2 | 1.3 ± 0.3 | 0 | 94.4 | 96 |
| | R286K | 2.5 ± 1.2 | 1.9 ± 0.5 | 32.3 | 45 | 92 |
| | H219Q | 1.2 | 1.1 | 25.2 | 23.3 | nd[c] |
| | G221E | 0.9 ± 0.07 | 3.6 | 0.06 | 0.06 | nd |
| | A326S | 0.7 ± 0.01 | 3.9 | 13 | 5.2 | nd |
| | I333V | 2.2 ± 0.7 | 1.7 | 0 | 0.06 | nd |
| | I376V | 1.9 ± 0.3 | 12 | 16.9 | 17.6 | nd |
| 3 | T332S/V370A | 1.9 ± 0.5 | nd | 0.06 | 3.1 | nd |
| | R286K/T332S | 2.7 ± 0.3 | nd | 0.25 | 7.9 | 106 |
| | R286K/A326T | 3 ± 0.3 | 3.7 ± 1.5 | 0 | 0.02 | 106 |
| | A326T/V370A | 3.4 ± 0.7 | 21 ± 6.0 | 0 | 0.02 | 93 |
| | R286K/V370A | 5.3 ± 0.1 | 381 ± 98 | 4.32 | 17.8 | 36 |
| | A326T/V362I | 12 ± 3.3 | 4.5 | 0 | 0 | 93 |
| | V362I/T332S | 5.7 ± 1.7 | 2.9 | 0 | 0.4 | 65 |
| | A326T/ΔV370 | 29 | 153 | 0 | 0 | nd |
| | R286K/V362I | 61 ± 35 | nd | 2.96 | 2.5 | 41 |
| | V362I/V370A | 208 ± 1.7 | >862 | 1.79 | 5.4 | 60 |
| 4 | R286K/A326T/V370A | 159 ± 123 | >233 | 0 | 0 | 86 |
| | H219Q/V362I/V370A | 357 ± 86 | nd | 0.31 | 1.1 | 115 |
| | R286K/A326T /V362I | 437 ± 142 | nd | 0 | 0 | 94 |
| | V362I/V370A/ΔT371 | 1072 ± 321 | >862 | 0 | 0 | nd |

[a]: 2018 version

[b]: NL4-3 virus

[c]: nd = not determined; RC, replicate capacity; FC EC$_{50}$: EC$_{50}$ target virus/EC$_{50}$ NL$_{4-3}$ wt virus; All experiments were performed in triplicate, and standard deviations were calculated from at least two separate experiments.

When an additional substitution was added to form triple combinations of substitutions (Group 4), greater decreases in susceptibility to GSK3532795 were observed, with fold changes of 159–1072. However, under these cell culture conditions, the secondary substitution H219Q rescued the reduced RC of V362I/V370A from 60% to 115% for the virus with the H219Q/V362I/V370A genotype (Table 4).

The differences in GSK3532795 susceptibility observed between the double V362I/V370A variant and the other V362I and V370A double and triple variants indicates that GSK3532795 susceptibility of V362I- and V370A-containing variants is context dependent. In general, double combinations of changes with V362I or V370A (Group 3) had more modest effects on susceptibility to GSK3532795, with fold changes of 1.9 to 12. However, the addition of R286K, particularly in the context of triple changes, produced larger effects on susceptibility.

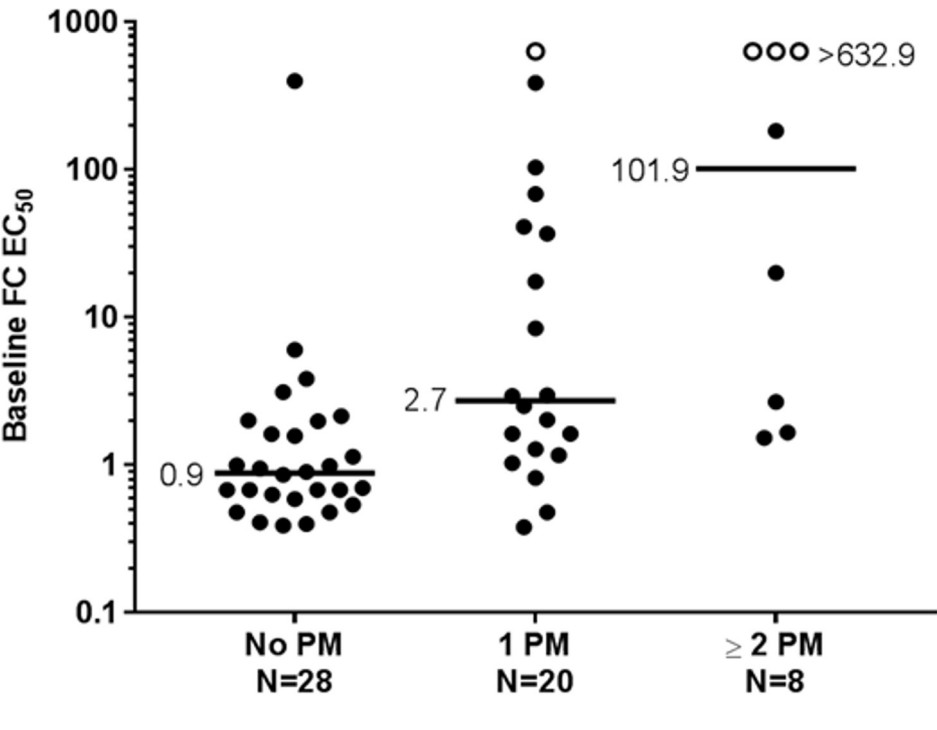

**Fig 2. Baseline GSK3532795 susceptibility relative to number of Gag polymorphisms in plasma-derived virus from patients included in part A of the phase 2a study (AI468002).** Baseline PMs included any change at V362, A364, Q369, V370, or T371. Black lines show the median for each group. FC-EC$_{50}$, fold-change in 50% effective concentration relative to reference virus; PM = polymorphism.

### Impact of baseline Gag polymorphisms on clinical responses to GSK3532795

Study AI468002 treated participants with doses of 5–120 mg of GSK3532795 once daily for 10 days as monotherapy and demonstrated potent antiviral activity (at 10 mg and above) against both subtype B and C viruses, although response was variable [20]. To correlate response with genotype/phenotype, samples from participants in Part A were analyzed at baseline for known MI and secondary polymorphisms, and after dosing for emergent substitutions (**Fig 2**). Genotypic analysis of baseline plasma HIV samples using the Gag sequencing assay at Monogram Biosciences was successful for 57/60 participants in Part A (subtype B infected participants). A subset of the Gag sequences of these participants are aligned and shown in **S2 Table**. These Gag genes were analyzed at Monogram Biosciences for the phenotype toward GSK3732795 (reported as FC IC$_{50}$ compared to a control Gag gene (HXB2)). This analysis identified at least one Gag polymorphism at positions 362, 364, 369, 370, 371 in 28 participants. Eight of these participants possessed 2 or more of these polymorphisms at baseline. **Fig 3** illustrates the susceptibility of the Gag/Pr phenotyped viruses to GSK3532795. Viruses without Gag polymorphisms had a median FC-IC$_{50}$ of 0.9-fold, with a range of 0.48–6.05, excluding one sample showing an unexplained FC-IC$_{50}$ of 398. Of note, this subject had a response to treatment with GSK3532795 at a dose of 20mg, with a viral load reduction of ~-1 log$_{10}$ copies/mL at Day 10. The 20 viruses with single Gag polymorphisms had a median FC-IC$_{50}$ of ~2.7. Of these, 12 exhibited GSK3532795 susceptibility comparable with viruses without Gag polymorphisms

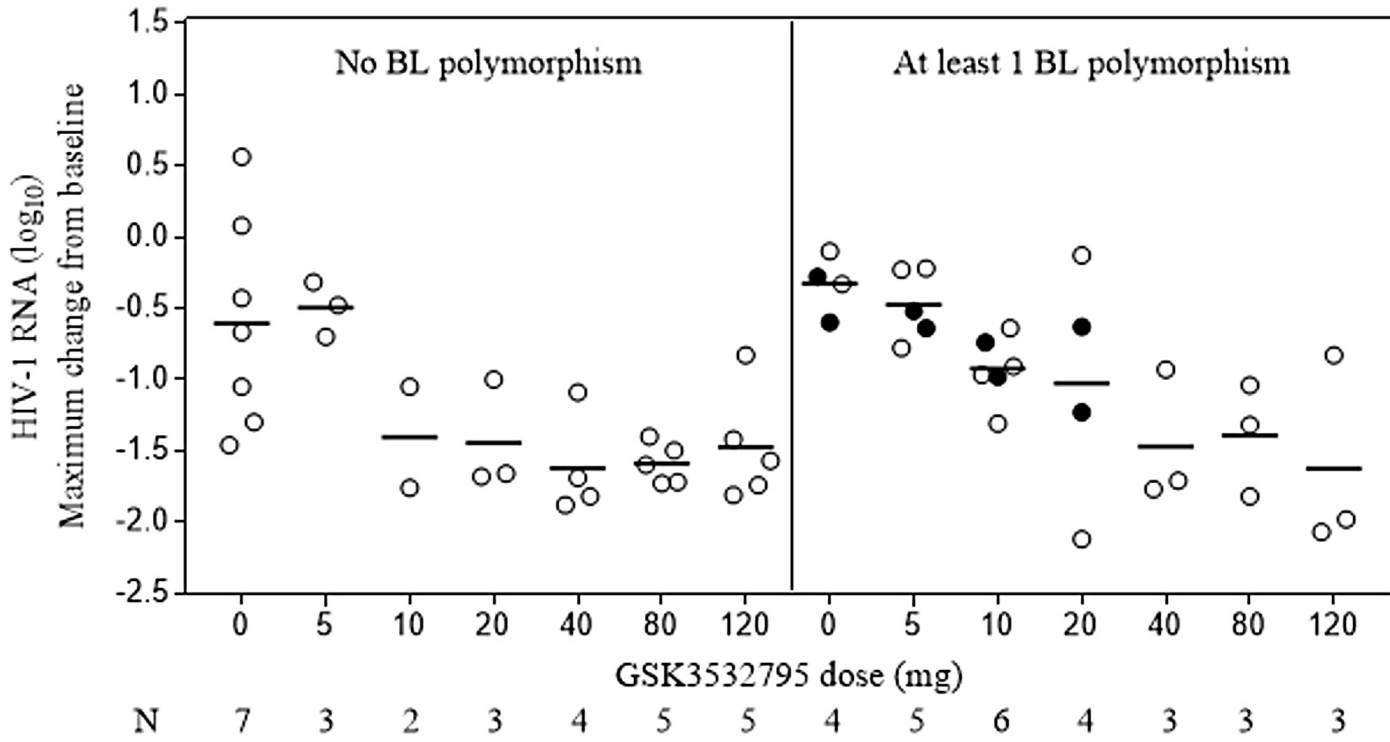

**Fig 3. Maximum viral load decline during 10-day monotherapy with GSK3532795 according to dose and detection of HIV-1.** Baseline polymorphisms included any change at V362, A364, Q369, or V370, unless otherwise indicated. Open circle: single polymorphism; closed circle: ≥2 polymorphisms; line: mean value of $\log_{10}$ HIV RNA.

(FC-IC$_{50}$ 0.38–2.98), and 8 showed decreased GSK3532795 susceptibility, with FC-IC$_{50}$ values 8.44- to >632-fold higher than wild-type control virus. The 8 samples with two or more Gag polymorphisms tended to have decreased susceptibility to GSK3532795, as FC-IC$_{50}$s ranged from 1.53 to >632 (with 3/8 having an FC-IC$_{50}$ >632). Analysis of the genotypes showed that three participants had polymorphisms at both positions 370 and 371 (one had undetermined amino acids at 370 and 371), two participants contained polymorphisms at positions 362 and 370, one participant had polymorphisms at 362 and 371, one participant had polymorphisms at 362, 370 and 371 and one participant contained polymorphisms at positions 362 and 369–371. (S2 Table).

## Genotypic and phenotypic changes in plasma HIV samples during treatment with GSK3532795

In Part A of the Phase 2a study (AI468002) matched baseline and on-treatment virus genotypic and phenotypic data were available for 37 participants in the GSK3532795 treatment groups, along with 4 participants from the placebo group. Analysis of emergent resistance was not performed for participants in the 5-mg cohort since there was no antiviral effect observed in this dose arm [20]. There were no genotypic (at positions 362–371) or phenotypic changes observed in the 4 participants from the placebo group. Virus genotypic changes (emergent or selected from mixtures at baseline) were seen during GSK3532795 treatment at Day 10 for 18 participants, across GSK3532795 doses from 10 to 120 mg (Table 5). Using a cut-off for phenotypic change as a >3-fold decrease in susceptibility from baseline to Day 10 to allow for assay variability, a total of 13 (of 37) participants had both emergent genotypic changes (or selection of a known polymorph from a mixture at baseline) together with phenotypic changes.

**Table 5. Baseline and day 10 on-treatment genotype and phenotype in the AI468002[a].**

| # | Dose (mg) | BL Gag[b] | Emergent or selected genotypic changes[c] | Day 0/10 EC$_{50}$ | Ratio day 10/day 0 FC EC$_{50}$s | VLR Day11 | VLR Max |
|---|---|---|---|---|---|---|---|
| 34 | 10 | WT | A364A/V | 0.630/42.2 | 67 | -1.16 | -1.76 |
| 13 | 20 | WT | V370I/V | 0.48/0.52 | 1.1 | -1.59 | -1.68 |
| 1 | 40 | WT | V362I/V, A364A/V | 0.39/3.79 | 9.7 | -0.95 | -1.09 |
| 44 | 40 | WT | V370M | 1.14/132 | 116 | -1.69 | -1.69 |
| 85 | 80 | WT | V362I/V, A364V/A | 3.13/146 | 47 | -1.31 | -1.5 |
| 98 | 80 | WT | A364V/A | 0.95/0.94 | 0.99 | -1.73 | -1.73 |
| 103 | 120 | R286K/R | R286K, V362I | 6.05/>704 | >116 | -0.83 | -0.83 |
| 111 | 120 | R286K | V370A/V | 3.85/15.8 | 4.1 | -1.53 | -1.57 |
| 17 | 10 | V370M | V362I/V | 2.52/1.57 | 0.62 | -1.02 | -1.31 |
| 25 | 10 | V362I/V | V362I | 41.1/423 | 10 | -0.9 | -0.97 |
| 26 | 20 | V370A T371N | V362I/V, A364A/V | 1.66/>666 | >401 | -0.64 | -1.23 |
| 52 | 20 | V370M | A364A/V | 0.480/106 | 221 | -1.94 | -2.12 |
| 22 | 40 | R286K, V370M | Q369Q/H | 1.63/20.3 | 12 | -0.93 | -0.93 |
| 38 | 40 | R286K/R, 370A/V | V370A | 68.5/496 | 7.2 | -0.64 | -1.71 |
| 109 | 120 | R286K, V362I/V | V362I | 2.95/407 | 138 | -0.81 | -0.83 |
| 110 | 120 | V370A/V | R286R/K, V370X[d], T371X | 0.82/5.2 | 6.3 | -1.54 | -2.07 |
| 43 | 10 | V362I, L363M, T371N | Δ371 | >632/>666 | 1 | 0.66 | -0.74 |
| 86 | 80 | V370M | A366A/V | 2.03/2.26 | 1.1 | -0.93 | -1.04 |

BL, baseline; FC, fold change relative to reference wild-type (NL4-3); EC50, 50% effective concentration; Viral load reduction in log$_{10}$ copies/m

[a]: Results are shown for participants where both baseline and on-treatment genotypic and phenotypic results were available

[b]: Changes either emerged new or were selected from a mixture at baseline

[c]: Baseline polymorphisms and newly emergent changes included any change at R286, V362, A364, A366, Q369, or V370

[d]: Sequence at 370, 371 could not be determined at Day 10 but was V370, T371 at Day 14.

Five other participants did not exhibit a significant phenotypic change (<3-fold) but did have an emergent genotypic change. In 4 of these 5 participants (# 13, 17, 86, 98), this amino acid was a mixture with wild type sequence, which is presumably the reason for the low phenotypic change. Additionally, 3 participants with mixtures at known polymorphic amino acids (#s 25, 38, 109) converted to a single amino acid polymorphism by Day 10, resulting in a significant (> 3 FC) phenotypic change from baseline. Another subject contained 3 polymorphisms at baseline and had a ΔT371 emerge by Day 10 (# 43). This subject had much reduced susceptibility to GSK3532795 at baseline (>632-FC), and a similar lack of sensitivity at Day 10. In addition, this subject had a suboptimal (<1 log$_{10}$ FC) response to treatment.

Although treatment with GSK3532795 ended on Day 10, an additional sample was obtained on Day 14 for genotypic analysis. Given the fact that numerous emergent substitutions occurred during drug treatment, it is not surprising that additional emergent mutations occurred during the days between the end of dosing and the Day 14 sampling when GSK3532795 plasma levels are decreasing to suboptimal levels due its long plasma half-life [14]. In all, 16 participants had 1 or more changes at Day 14 compared to Day 10 (at one or more amino acid positions 362, 364, 369–371), with 10 and 6 participants having A364A/V or V362V/I emerge (one participant had both emerge), respectively. In addition, 2 participants (#s 2, 11) had V370V/L or V370M emerge and 1 participant (#106) had an emergent A366A/V (the latter along with an emergent A364A/V). In addition, V362V/I (2 participants, #s 1, 17) and V370V/I (participant # 13) or V370M (participant # 44) were lost in the Day 14 sample compared to the Day 10

sample from those participants. The only other change in a Day 14 sample was a reversion back to T371N (T371N was in the Baseline sample, ΔT371 was in the Day 10 sample) in one participant (#43) (S2 Table). In 6 (of the 15 participants, #34, 86, 90, 106,112 and118), the FC-IC$_{50}$s were significantly (>3-fold) higher at Day 14 (versus Day 10), with 5/6 coming from the higher dose groups (80 mg, 2 participants; 120 mg, 3 participants). In addition, one participant who lost a V370M mutation at Day 14 (#44; 40 mg dose group), susceptibility was significantly increased with a decrease in FC-IC$_{50}$ from 132 at Day 10 to 0.48 at Day 14.

For the most part, the emergent changes observed during treatment (Table 5) mirrored those observed during *in vitro* selections (Tables 1–3). Mixed populations of substitutions tended to be associated with smaller changes in GSK35323795 susceptibility than populations containing known MI substitutions. In participants where Gag mutations were selected in addition to existing baseline polymorphisms, the baseline polymorphisms were either R286K or V370M/I or V370A. In the cases where phenotypic changes were seen in the absence of new treatment-emergent Gag substitutions, the baseline genotype showed a mixed population of Gag polymorphisms at positions V362 or V370 that was consolidated during treatment, for example, mixed V362V/I converting to V362I. Other emergent substitutions in the SP1 region included Q369Q/H in one subject (#22; 40 mg dose) and Δ371 in another (#43; 10 mg dose). In terms of known secondary polymorphisms, there were no observed changes in any of these, including R286, except for H219. One subject had H219H/P and another 2 had H219H/Q emerge at Day 10, with all 3 reverting back to H219 at Day 14. In contrast to *in vitro* evaluation of H219 variant viruses in laboratory strains, 219 polymorphic changes in clinical samples were not associated with altered FC-IC$_{50}$s or altered VLR. Two other participants had either H219H/P or H219H/Q at baseline, but also reverted to H219 at Day 10. The protease regions of viruses at baseline, Day 10 and Day 14, were also sequenced. In no cases were there emergent substitutions observed in protease in these GSK33279 treated participants.

## Structural model of bound GSK3532795

A structural model of the CA/SP1 hexamer/GSK3532795 complex is shown in Fig 4 indicating the locations of a subset of the residues described above. In the model, GSK3532795 is bound within the six-helix bundle formed from C-terminal residues of the CA CTD and part of the SP1 region. Multiple hydrophobic interactions between the ligand and the L363, M367, V370 and T371 side chains are observed. In addition, the ligand carboxylate moiety is proximal to the side chains from a ring of six basic K359 residues and could provide favorable salt bridge interactions. Overall, the modeled binding mode is similar to that suggested by Purdy et al. for bevirimat [27]. These protein-ligand contacts in the complex may provide additional stabilization to the CA/SP1 six helix bundle, thus reducing the fraction of protein in which the peptide including the CA/SP1 cleavage site is in an extended conformation necessary for cleavage by HIV protease [35]. This stabilization and resultant reduction in the rate and/or extent of CA/SP1 cleavage could negatively impact the viral maturation process. This represents a likely mechanism of action for bevirimat and potentially for GSK3532795 as well [21, 27, 36, 37].

Some of the mutations conferring resistance to maturation inhibitors are thought to act by destabilizing the CA/SP1 assembly to counteract the stabilization derived from the presence of bound MI, though other mechanisms are possible as summarized by Urano et al. [21, 38] Primary resistance mutation residues V362, A364, and secondary mutation site V370 are located in the CA/SP1 six-helix bundle, but only the latter forms any direct side chain contacts with GSK3532795. Based on the model, the 835-fold resistance observed for the A364V mutant is unlikely to arise from unfavorable contacts between the ligand and the residue 364 side chain in the mutant protein. As shown in Fig 5, the A364 side chain on each monomer in the CA/

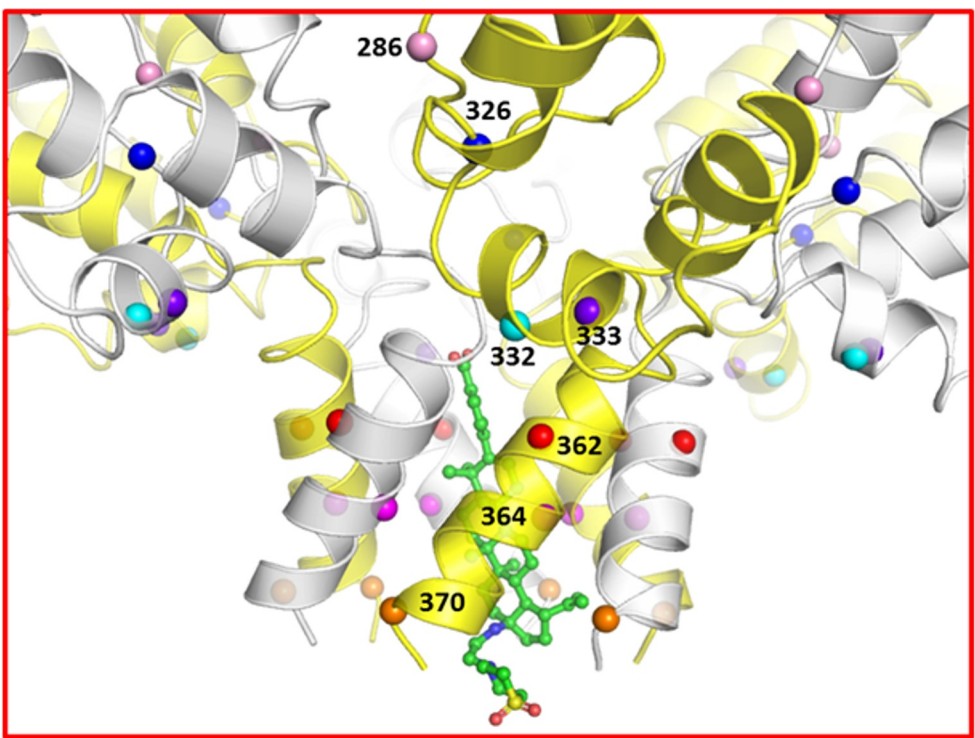

**Fig 4. Model of the CA-SP1/GSK3532795 complex.** The protein is depicted in a white and yellow semi-transparent cartoon representation with the locations of residues 286, 326, 332, 333, 362, 364, and 370 highlighted with pink, blue, cyan, purple, red, magenta, and orange spheres, respectively. GSK3532795 is shown in ball and stick representation with green carbon atoms. Image created with PyMol (2.1.1, Schrödinger, LLC).

SP1 six helix bundle occupies a small hydrophobic pocket formed in part by the side chains of the A366, M367, and V370 residues on an adjacent CA/SP1 monomer and is not involved in direct contacts with the bound ligand. There is limited volume in this pocket, and replacement of the alanine side chain methyl group in the WT protein with the larger isopropyl moiety in the A364V resistance mutant may destabilize the six-helix bundle by preventing optimal engagement of the adjacent CA/SP1 monomers via steric blockade, thus increasing the rate of HIV protease cleavage and the rate of dissociation of bound GSK795, as previously reported [16]. Alternatively, the A364V mutation could modify the conformation of the nearby M367 side chain in a manner unfavorable for GSK3532795 binding.

As shown (**Table 4**), the impact of the V362I mutation alone on GSK3532795 susceptibility was minimal, although combinations with secondary mutations R286, A326 or T332, resulted in higher levels of resistance. This suggests that any structural changes induced by the single V362I mutation are likely to be small. **Fig 5** provides a detailed view of V362 and proximal residues in the CA/SP1 model. Each V362 side chain occupies a pocket formed by H358, R361, E365, and A366 on the same monomer and P356, G357, and A360 on the adjacent CA/SP1 monomer and makes substantial van der Waals contacts with G357, A360, and H358. There appears to be volume available to accommodate the larger isoleucine side chain in the V362I mutant protein. However, it is interesting to note the proximity of V362 to both helix 10 and a β-turn (residues 352–356) in the CA CTD, which are believed to stabilize the immature capsid structure [21, 36]. D329, located at the N-terminal end of helix 10 is thought to stabilize the CA/SP1 assembly through interactions with H358 on the same CA/SP1 monomer, and P356 on an adjacent CA/SP1 monomer in the hexameric assembly [21]. Alanine mutation of both

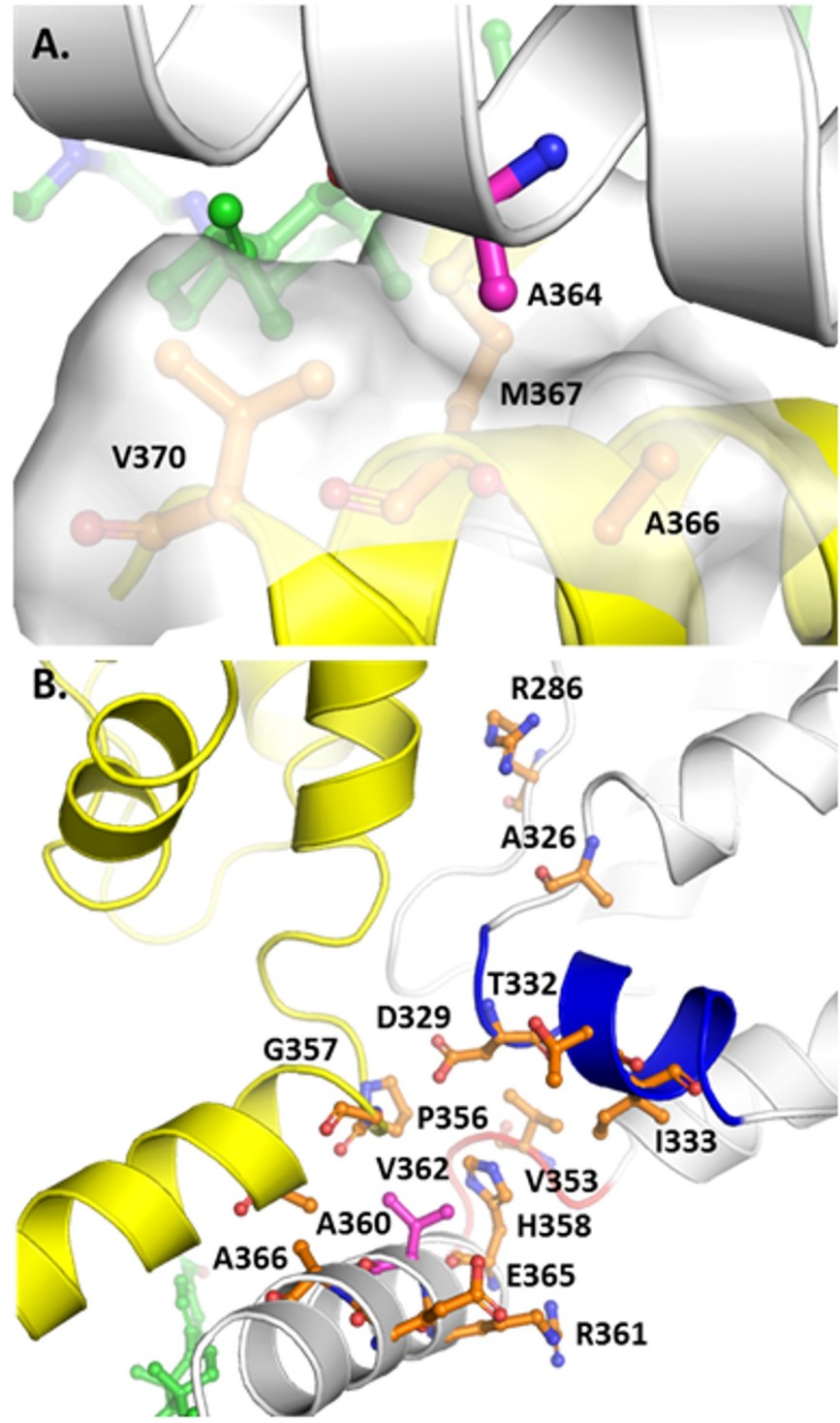

**Fig 5. Protein environments of A364 (A, top) and V362 (B, bottom) in the CA-SP1/GSK3532795 complex model.**
The protein backbone is depicted in white and yellow cartoon representation with helix 10 and the 352–356 β-turn colored blue and red, respectively. GSK3532795 and specific residues are shown in ball and stick style with carbon atoms colored magenta (A364, V362), green (GSK3532795), or orange. Image created with PyMol (2.1.1, Schrödinger, LLC).

D329 and residue V353 in the 352–356 β-turn resulted in either aberrant or mature pheno-
types for HIV-1 ΔMA-CA-SP1-NC Gag when DNA or tartrate, respectively, were used to stim-
ulate in vitro assembly, while the WT protein assembles into immature-like virus like particles
under the same conditions, suggesting the importance of these residues for immature capsid
stability [36]. Two of the three CA CTD secondary mutations (T332S, A326T) are either
located in helix 10 (T332) or nearly within van der Waals contact distance of helix 10 residues
(A326). It is plausible that the combination of the V362I mutation with changes at residues
326 or 332 negatively impacts interactions involving helix 10 and/or the proximal 352–356 β-
turn, which serve to stabilize the immature CA/SP1 assembly.

Secondary resistance polymorph R286K is somewhat more distant from V362. In the con-
strained molecular dynamics simulation of the CA/SP1 hexamer/GSK3532795 complex, the
R286 side chain in each CA/SP1 monomer was observed to form frequent hydrogen bonds
with Q351 and T348 and salt bridges with E344 on an adjacent CA/SP1 monomer. It is possi-
ble that these interactions contribute to the stability of the hexameric CA/SP1 assembly. These
could be diminished slightly in the R286K mutant, but not enough to dramatically impact the
overall stability of the CA/SP1 helix in the absence of additional destabilizing mutations.

Secondary resistance polymorph R286K is somewhat more distant from V362. In the con-
strained molecular dynamics simulation of the CA/SP1 hexamer/GSK3532795 complex, the
R286 side chain in each CA/SP1 monomer was observed to form frequent hydrogen bonds
with Q351 and T348 and salt bridges with E344 on an adjacent CA/SP1 monomer. It is possi-
ble that these interactions contribute to the stability of the hexameric CA/SP1 assembly. These
could be diminished slightly in the R286K mutant, but not enough to dramatically impact the
overall stability of the CA/SP1 helix in the absence of additional destabilizing mutations.

## Discussion

We previously reported on the *in vitro* activity of GSK3532795 (mean $EC_{50}$ = 3.9 nM) against a
library of 87 gag/pr recombinant viruses representing 96.5% of the subtype B polymorphic Gag
diversity near the CA/SP1 cleavage site [15]. The study showed that GSK3532795 exhibited a
broad spectrum of potent activity against a wide range of HIV-1 viruses. However, it did not
directly address resistance to this compound. This study was carried out to understand the *in vitro*
resistance profile and durability of GSK3532795 and to correlate it with the clinical response pro-
file and emergent substitutions observed during a 10-day monotherapy study with GSK3532795.

Selection for viruses with decreased susceptibility to GSK3532795 used two alternate meth-
ods. One used selection via serial passage at increasing concentrations of compound, while the
other used passage at a high fixed dose ($30 \times EC_{50}$) of GSK3532795. In addition, viruses with dif-
ferent polymorphisms in the SP1 region were used in some of these experiments. The results
of all these experiments produced similar resistance substitutions, with A364V and V362I
being predominantly selected, although the V362I mutation needs to be present in conjunc-
tion with other polymorphisms, such as V370A, in order to greatly reduce susceptibility to
GSK3732795 in vitro. These substitutions are close to (V362I) or adjacent to (A364V) the
HIV-1 protease cleavage site (L363/A364) that is inhibited by the action of GSK3532795.
A364V rarely is found in the LANL database in any subtype, however, *in vitro* selection for
bevirimat resistance from wild-type virus yielded this substitution,[34] and A364V was also
reported in two HIV-1-infected participants in a bevirimat clinical trial [39]. In our experi-
ments, virus populations that acquired A364V had greatly reduced GSK3532795 susceptibility
and this was confirmed by the single A364V site-directed mutant, which had 835-fold reduced
GSK3532795 susceptibility but maintained a RC of 82%. V362I is a polymorphism in subtype
B isolates (15.4% in the LANL database), and, depending on background, is associated with

bevirimat resistance [9, 10, 34]. This substitution was generally associated with smaller reductions in GSK3532795 susceptibility and the single V362I site-directed mutant remained susceptible to GSK3532795. The impact of this substitution appears to be dependent on the context of secondary substitutions. Although neither V362I nor V370A single changes significantly altered susceptibility to GSK3532795 (2-fold for each virus), the V362I/V370A combination resulted in a 208-fold reduction in GSK3532795 sensitivity. In two *in vitro* passage experiments, the V362I was replaced with A364V, conferring a larger reduction in GSK3532795 susceptibility and better replication and suggesting that V362I alone in Gag produces an unfit virus. This correlates with the low replication capacity (60%) of the double mutant virus.

In addition to V362I and A364V, a number of secondary mutations were selected. Secondary substitutions selected *in vitro* mapped to three locations, the CA-CTD (R286K, A326T, T332S and I333V), the cyclophilin A (CypA) binding domain of Gag (V218/A/M, H219Q and G221E), and protease (R41G). Site directed mutagenesis showed that none of these secondary substitutions in isolation had a significant impact on GSK3532795 susceptibility. An example in this regard is R286K, which by itself only showed a 2.5-FC in a site-directed mutant, while a double mutant with V362I induced a 61-FC. Further, addition of A326T to the duo induced a 437-FC (**Table 4**). Of note, subtype B contains 4.3% of the R286K/V370A double polymorphism and in the clinical study, one subject with a baseline R286K (FC 3.9) acquired a V370A/ V mixture during dosing and displayed emergent resistance on-therapy (FC 15.8). *In vitro*, the addition of T332S, A326T or R286K to V362I increased the fold-reduction in GSK3532795 $EC_{50}$ from 2.2 to 5.7, 12, and 61-fold, respectively. The structural model of the CA/SP1 hexamer suggests a potential hexamer stabilizing role for R286 through inter-CA/SP1 monomer hydrogen bonding and/or salt bridge formation which may be negatively impacted by mutation to Lys. A326 and T332 are located within or proximal to CA CTD helix 10 and/or the 352–356 β-turn which appear to be involved in CA/SP1 helix stability. Both helix 10 and the β-turn are close to residue 362 in the model, and mutations of A326 or T332, in combination with the V362I mutation may serve to destabilize the immature CA/SP1 assembly.

Substitutions at positions 218 and 219 in the CypA binding loop are known to confer replication advantages in certain genotypic backgrounds as a function of cell type, possibly by reducing intra-virion CypA incorporation [40, 41]. H219Q was another secondary mutation observed *in vitro*. H219Q is found in the CypA binding loop and was shown in to increase the RC of the poorly growing V362I/V370A virus from 60% to 115%. The clinical significance of the CypA changes outside of the laboratory strain of HIV-1 used in the selection experiments ($NL_{4-3}$), and the cells used to propagate the virus (MT-2) is unknown, especially given the wide range of CypA levels present in different cell types, and the complex interplay between viral assembly and CypA levels [40–43]. In contrast to *in vitro* selections, baseline H219 polymorphisms or emergent changes at amino acid Gag 219, did not appear correlated with either GSK3532795 susceptibility or VLR, thus suggesting that susceptibility to 219 changes is an in vitro phenomenon.

R41G emerged alongside V362I/T332S *in vitro* and was shown to increase the GSK3532795 resistance of this double variant from 5.7- to 217-fold. R41G is present in 1/3083 isolates in the LANL database and is not a primary PI resistance substitution [11, 44]. A related change, R41K, is a common polymorph (26% in subtype B and 68% of all Gag genes in the LANL database), and may be involved in the emergence of protease resistance to an investigational protease inhibitor [45, 46]. R41 is located in a loop proximal to the HIV-1 protease substrate binding site and might act allosterically to facilitate closing the pocket over the substrate, thereby allowing catalysis. R41G might alter the dynamics of the loop motion and the final positioning of the loop, which could cause the active site to better recognize V362I variants such as V362I/T332S. It is interesting to note that no emergent protease mutations were observed in the GSK3532795 clinical study.

As previously noted, double polymorphic viruses such as V362I/V370A were selected *in vitro* and observed in the clinic. As reported for A364V [16], its relative rate of cleavage at CA/SP1 by HIV-1 protease was 9.7-fold elevated vs wild type virus. Similarly, V362I/V370A also demonstrated an elevated cleavage rate (4.9-fold) and both viruses showed less than 100% maximal inhibition (MPI) by GSK3532795. Overall, these data argue for elevated protease cleavage rates as correlative with both incomplete inhibition and elevated FC-$IC_{50}$ values and together provide a basic mechanistic understanding for the development of resistance to GSK3532795.

In the Phase 2a clinical study at daily GSK3532795 doses of 40–120 mg, similar median maximum changes in HIV-1 RNA were seen in the presence or absence of Gag polymorphisms associated with bevirimat resistance [20]. Similar to what was observed *in vitro*, doses of 10–120 mg of GSK3732795 primarily selected at Day 10 for emergent substitutions (or selection from a mixture at baseline) at 2 positions, V362I (N = 7/18 participants) and A364V (N = 6/18), with 3 of these participants (#s 1, 26 and 80) selecting both changes. Interestingly, one subject (# 22) with baseline R286K/V370M and BL FC of 1.6 acquired the Q369Q/H mixture on dosing, displayed a reduced susceptibility at Day 11 compared to baseline (FC = 12) and exhibited a virologic response of -0.93 $\log_{10}$ RNA. Another participant (# 44) started out with a wild type genotype but had an emergent V370M on Day 11 with a FC = 116. However, this participant reverted to V370 by Day 14. A V370M containing virus was previously shown to be sensitive to GSK3532795 (FC = 2.8 [15]), so it is not surprising that this participant had a good clinical response (viral load reduction = -1.69 $\log_{10}$). All the emergent substitutions observed in this study were similar to a subset observed during *in vitro* selections with this compound and correlated to similar studies with bevirimat, suggesting the relevance of *in vitro* selection studies for evaluation of the resistance profile of maturation inhibitors.

The monotherapy study with GSK3532795 in HIV-1 infected participants showed that at doses of 10–120 mg once daily, a large proportion of the participants selected (either as emergent or from a baseline mixture) for substitutions known to decrease susceptibility to the compound (Table 5). This suggests that the resistance barrier to GSK3532795 is relatively low. This correlates with the results from the Phase 2b study, which dosed HIV-1 infected participants with 60, 120 or 180 mg GSK35323795 QD along with TDF/FTC (300/200 mg) [19]. Although efficacy rates of these arms with the comparator (EFV with TDF/FTC) were similar, the GSK3532795 arms showed a higher rate of treatment-emergent resistance to the NRTI backbone than the EFV control arm. That data, along with issues with gastrointestinal intolerability in the Phase 2b study, led to the termination of the development program for GSK3532795 [19]. Improvements in safety and resistance barrier may be needed for a next generation maturation inhibitor to succeed in development.

## Supporting information

**S1 Table. Amino acid substitutions selected in gag during independent sequential passage experiments in the presence of increasing concentrations of GSK35323795.**
(PDF)

**S2 Table. Sequences of key amino acids in Gag in subjects treated with 5–120 mg GSK3532795.**
(PDF)

## Acknowledgments

The manuscript is dedicated to the memory of Beata Nowicka-Sans, a great scientist, collaborator and friend, who performed many of the studies described here. A subset of the results

presented here were presented at the 9th IAS Conference on HIV Science; July 23–26, 2017, Abstract 347; Resistance Profile of HIV-1 Maturation Inhibitor GSK3532795. Paris, France.

## Author Contributions

**Conceptualization:** Ira Dicker, Neelanjana Ray, Brett R. Beno, Alicia Regueiro-Ren, Samit Joshi, Mark Krystal, Max Lataillade.

**Formal analysis:** Ira Dicker, Neelanjana Ray, Brett R. Beno, Alicia Regueiro-Ren, Mark Krystal, Max Lataillade.

**Investigation:** Sharon Zhang, Samit Joshi, Max Lataillade.

**Methodology:** Alicia Regueiro-Ren, Samit Joshi, Max Lataillade.

**Supervision:** Ira Dicker, Mark Cockett, Mark Krystal, Max Lataillade.

**Writing – original draft:** Ira Dicker.

**Writing – review & editing:** Ira Dicker, Neelanjana Ray, Brett R. Beno, Alicia Regueiro-Ren, Samit Joshi, Mark Cockett, Mark Krystal, Max Lataillade.

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
