## [Decision Letter · Decision Letter 0]

20 Sep 2019

PONE-D-19-22219

Resistance profile of the HIV-1 maturation inhibitor GSK3532795 in vitro and in a clinical study

PLOS ONE

Dear Dr. Krystal,

Thank you for submitting your manuscript to PLOS ONE. After careful consideration, we feel that it has merit but does not fully meet PLOS ONE’s publication criteria as it currently stands. Therefore, we invite you to submit a revised version of the manuscript that addresses the points raised during the review process.

While this is generally a well performed study a major concern is that EC50 values lack a meaure of variance (i.e. SEM), it is not clear how many independent assays were peformed to reach these values, and  no statistical analyses have been performed.

We would appreciate receiving your revised manuscript by Nov 04 2019 11:59PM. To enhance the reproducibility of your results, we recommend that if applicable you deposit your laboratory protocols in protocols.io, where a protocol can be assigned its own identifier (DOI) such that it can be cited independently in the future. For instructions see: http://journals.plos.org/plosone/s/submission-guidelines#loc-laboratory-protocols

We look forward to receiving your revised manuscript.

Kind regards,

Gilda Tachedjian, Ph.D.

Academic Editor

PLOS ONE

Journal Requirements:

"All co-authors are or were employees of Bristol-Myers Squibb (the sponsor at that time) at the time this work was performed."

We note that one or more of the authors are employed by a commercial company:ViiV Healthcare and Bristol-Myers Squibb

Reviewers' comments:

Reviewer's Responses to Questions

**Comments to the Author**

1. Is the manuscript technically sound, and do the data support the conclusions?

Reviewer #1: Yes

Reviewer #2: Yes

2. Has the statistical analysis been performed appropriately and rigorously? 

Reviewer #1: Yes

Reviewer #2: No

3. Have the authors made all data underlying the findings in their manuscript fully available?

Reviewer #1: Yes

Reviewer #2: Yes

4. Is the manuscript presented in an intelligible fashion and written in standard English?

Reviewer #1: Yes

Reviewer #2: Yes

5. Review Comments to the Author

Reviewer #1: Reviewer comments:

GSK3532795 (formerly BMS955176) is a second-generation maturation inhibitor (MI) that has advanced to Phase 2b clinical trial for treatment of HIV-1 infection but discontinued from further development. In this study, resistance development to GSK3532795 was evaluated through in vitro methods by using various strategies. The reduced susceptibility to GSK3532795 mapped specifically to amino acids near the capsid/ spacer peptide 1 (SP1) junction, the cleavage of which is blocked by MIs. Two key substitutions, A364V or V362I, were selected, along with few other secondary substitutions. These mutations were similar to the resistance profile of BVM. Mutations acquired in in vitro experiments were then correlated with information obtained in a Phase 2a proof-of-concept study in HIV-1 infected participants. In the Phase 2a study, a subset of these substitutions was also observed at baseline and some were selected following GSK35323795 treatment in HIV-1-infected participants. All the emergent substitutions observed in this study were similar to a subset observed during in vitro selections with this compound. This study suggested the relevance of in vitro selection studies for evaluation of the resistance profile of maturation inhibitors. Furthermore, in this study, they structurally modelled the compound with immature capsid hexamer which suggested possible mechanism of action and interaction of MIs and capsid protein was similar for Bevirimat as well as GSK3532795. This is a well carried out study. I have the following suggestions:

Comments:

1. Abstract: Line no. 36: According to the present study, H219Q increased viral replication capacity and reduced susceptibility of poorly growing viruses “only in in vitro studies”. (Reference required)

2. Introduction: Line no. 57: Mention the studies done to identify other BVM resistant mutations (Reference required).

3. Introduction: Line no. 58: Add another reference Waki et.al, 2012, as they have identified more resistant mutants in presence of the same maturation inhibitor.

4. Introduction: Line no. 61: Reference no. 10 is not relevant in this case.

5. Materials and methods: Line no. 94-100: Please mention about the no drug control used in the study.

6. Materials and methods: Line no. 166: Please give full form of QD.

7. Materials and methods: Line no. 166: Only part A of AI68002 clinical study was used in this study. Hence there is no need to mention about part C. If it has to be mentioned here, please justify why part C was not used in the present study.

8. Results: Page no. 10: Along with the table 1, a pictorial representation of the sequence of CA-SP1 region which shows locations of mutated amino acids will help in better understanding of mutation pattern.

9. Results: Line no. 224: Reference is missing.

10. Results: Page no. 13: Along with the table 2, a pictorial representation of the sequence of CA-SP1 region which shows locations of mutated amino acids will help in better understanding of mutation pattern.

11. Results: Line no: 240-247: The paragraph reads rephrasing…

12. Results: Line no. 258: please explain the meaning of square?

13. Results: Line no 258: In the WT virus at amino acid position 371, there is T instead of V as mentioned. Please explain

14. Results: Line no. 264: Same as comment no. 13.

15. Results: Page no. 15: Along with the table 3, a pictorial representation of the sequence of CA-SP1 region which shows locations of mutated amino acids will help in better understanding of mutation pattern.

16. Results: Line no 275: Please add reference of Adamson et.al, 2006 also, as this work is also relevant.

17. Results: Line no. 287: Please add reference for the statistics given.

18. Results: Line no 289-290: Please rewrite the sentence.

19. Results: Line no. 303-304: The sentence is not clear.

20. Results: Line no. 333-339: It is clearly mentioned that 8 samples contains two or more Gag polymorphisms. But from the sentence “Analysis of the genotypes showed that 5 participants had polymorphisms at both positions 370 and 371 (one had undetermined amino acids at 370 and 371), two participants contained polymorphisms at positions 362 and 370, one participant had polymorphisms at 362, 370 and 371 and one participant contained polymorphisms at positions 362 and 369-371.”, the total patients become 9. Please check it carefully.

Reviewer #2: This manuscript describes the genotypes of GSK3532795 resistant viruses selected in vitro and in vivo. Although most of the findings are reminiscent of the prototypic HIV-1 maturation inhibitor Bevirimat, this work confirms the sites of main drug resistant mutations and the utility of accessory mutations in heighten the drug resistance. The strength of this work lies in a comprehensive characterization of the genotypes of drug resistant viruses derived from patient samples. This allows the authors to suggest that, for the most part, the emergent in vivo drug resistant mutations mirrored those observed from the drug resistant viruses selected in vitro.

The main weakness of this manuscript is that most data, such as the IC50s, were not processed/presented in any form of statistical analysis. Without detailed statistical methods/analyses, it is difficult for readers to evaluate the validity of the results and the rigor of the research.

6. PLOS authors have the option to publish the peer review history of their article (what does this mean?). If published, this will include your full peer review and any attached files.

Reviewer #1: No

Reviewer #2: No

---

## [Author Response · Author response to Decision Letter 0]

1 Oct 2019

Response to Reviewer’s Comments

PONE-D-19-22219

Resistance profile of the HIV-1 maturation inhibitor GSK3532795 in vitro and in a clinical study

PLOS ONE

Please see our response to the reviewer’s comments below:

Editor Comments:

5. Review Comments to the Author

Reviewer #1: Reviewer comments:

GSK3532795 (formerly BMS955176) is a second-generation maturation inhibitor (MI) that has advanced to Phase 2b clinical trial for treatment of HIV-1 infection but discontinued from further development. In this study, resistance development to GSK3532795 was evaluated through in vitro methods by using various strategies. The reduced susceptibility to GSK3532795 mapped specifically to amino acids near the capsid/ spacer peptide 1 (SP1) junction, the cleavage of which is blocked by MIs. Two key substitutions, A364V or V362I, were selected, along with few other secondary substitutions. These mutations were similar to the resistance profile of BVM. Mutations acquired in in vitro experiments were then correlated with information obtained in a Phase 2a proof-of-concept study in HIV-1 infected participants. In the Phase 2a study, a subset of these substitutions was also observed at baseline and some were selected following GSK35323795 treatment in HIV-1-infected participants. All the emergent substitutions observed in this study were similar to a subset observed during in vitro selections with this compound. This study suggested the relevance of in vitro selection studies for evaluation of the resistance profile of maturation inhibitors. Furthermore, in this study, they structurally modelled the compound with immature capsid hexamer which suggested possible mechanism of action and interaction of MIs and capsid protein was similar for Bevirimat as well as GSK3532795. This is a well carried out study. I have the following suggestions:

Comments:

1. Abstract: Line no. 36: According to the present study, H219Q increased viral replication capacity and reduced susceptibility of poorly growing viruses “only in in vitro studies”. (Reference required)

The abstract describes data in the manuscript and does not reference any prior studies. This is data included and discussed in the results section (Line 302 and Table 4). The authors do not feel that a reference is required here.

2. Introduction: Line no. 57: Mention the studies done to identify other BVM resistant mutations (Reference required).

Thank you for this comment. A number of these studies are referenced on line

3. Introduction: Line no. 58: Add another reference Waki et.al, 2012, as they have identified more resistant mutants in presence of the same maturation inhibitor.

Thank you. This reference has been added. 

4. Introduction: Line no. 61: Reference no. 10 is not relevant in this case.

Thank you. This reference was inadvertently added here and is now removed.

5. Materials and methods: Line no. 94-100: Please mention about the no drug control used in the study.

A sentence to this effect was added on Line 108-9. 

6. Materials and methods: Line no. 166: Please give full form of QD.

The definition of QD has been added on Line 177.

7. Materials and methods: Line no. 166: Only part A of AI68002 clinical study was used in this study. Hence there is no need to mention about part C. If it has to be mentioned here, please justify why part C was not used in the present study.

Thank you. This sentence has been removed.

8. Results: Page no. 10: Along with the table 1, a pictorial representation of the sequence of CA-SP1 region which shows locations of mutated amino acids will help in better understanding of mutation pattern.

A new Figure 1 was added as a pictorial representation of the protein region and bevirimat resistant mutations. This should address a few of this reviewer’s comments.

9. Results: Line no. 224: Reference is missing.

Thank you. This was a formatting error that occurred during pdf formation. The reference has been added back.

10. Results: Page no. 13: Along with the table 2, a pictorial representation of the sequence of CA-SP1 region which shows locations of mutated amino acids will help in better understanding of mutation pattern.

The new Figure 1 should address this.

11. Results: Line no: 240-247: The paragraph reads rephrasing…

The paragraph (Lines 250-258) has been rewritten for clarity.

12. Results: Line no. 258: please explain the meaning of square?

Again, this was a formatting error. It has been corrected.

13. Results: Line no 258: In the WT virus at amino acid position 371, there is T instead of V as mentioned. Please explain

This was an error and has been fixed. Amino acid 371 is a T. 

14. Results: Line no. 264: Same as comment no. 13.

Fixed as requested.

15. Results: Page no. 15: Along with the table 3, a pictorial representation of the sequence of CA-SP1 region which shows locations of mutated amino acids will help in better understanding of mutation pattern.

Again, the graphics in Figure 1 should cover this. 

16. Results: Line no 275: Please add reference of Adamson et.al, 2006 also, as this work is also relevant.

Adamson reference has been added.

17. Results: Line no. 287: Please add reference for the statistics given.

The reference was included earlier in the manuscript and has been referenced again here.

18. Results: Line no 289-290: Please rewrite the sentence.

Sentence has been rewritten for clarity (Lines 300-1)

19. Results: Line no. 303-304: The sentence is not clear.

Sentence has been rewritten for clarity (Lines 314-18)

20. Results: Line no. 333-339: It is clearly mentioned that 8 samples contains two or more Gag polymorphisms. But from the sentence “Analysis of the genotypes showed that 5 participants had polymorphisms at both positions 370 and 371 (one had undetermined amino acids at 370 and 371), two participants contained polymorphisms at positions 362 and 370, one participant had polymorphisms at 362, 370 and 371 and one participant contained polymorphisms at positions 362 and 369-371.”, the total patients become 9. Please check it carefully.

Thank you. There was an stated error that has been corrected in this version.

Reviewer #2: This manuscript describes the genotypes of GSK3532795 resistant viruses selected in vitro and in vivo. Although most of the findings are reminiscent of the prototypic HIV-1 maturation inhibitor Bevirimat, this work confirms the sites of main drug resistant mutations and the utility of accessory mutations in heighten the drug resistance. The strength of this work lies in a comprehensive characterization of the genotypes of drug resistant viruses derived from patient samples. This allows the authors to suggest that, for the most part, the emergent in vivo drug resistant mutations mirrored those observed from the drug resistant viruses selected in vitro.

The main weakness of this manuscript is that most data, such as the IC50s, were not processed/presented in any form of statistical analysis. Without detailed statistical methods/analyses, it is difficult for readers to evaluate the validity of the results and the rigor of the research.

Thank you for the comment. Statistical analyses have now been applied to the data in Table 4.

---

## [Editor Report · Decision Letter 1]

7 Oct 2019

Resistance profile of the HIV-1 maturation inhibitor GSK3532795 in vitro and in a clinical study

PONE-D-19-22219R1

Dear Dr. Krystal,

We are pleased to inform you that your manuscript has been judged scientifically suitable for publication and will be formally accepted for publication once it complies with all outstanding technical requirements.

With kind regards,

Gilda Tachedjian, Ph.D.

Academic Editor

PLOS ONE
---

## [Editor Report · Acceptance letter]

10 Oct 2019

PONE-D-19-22219R1 

Resistance profile of the HIV-1 maturation inhibitor GSK3532795 in vitro and in a clinical study 

Dear Dr. Krystal:

I am pleased to inform you that your manuscript has been deemed suitable for publication in PLOS ONE. Congratulations! Your manuscript is now with our production department. 

With kind regards,

on behalf of

Professor Gilda Tachedjian 

Academic Editor

PLOS ONE